# Computational Modelling and Sustainable Synthesis of a Highly Selective Electrochemical MIP-Based Sensor for Citalopram Detection

**DOI:** 10.3390/molecules27103315

**Published:** 2022-05-21

**Authors:** Patrícia Rebelo, João G. Pacheco, Iuliia V. Voroshylova, Isabel Seguro, Maria Natália D. S. Cordeiro, Cristina Delerue-Matos

**Affiliations:** 1REQUIMTE, LAQV, Instituto Superior de Engenharia do Porto, Instituto Politécnico do Porto, Rua Dr. António Bernardino de Almeida 431, 4200-072 Porto, Portugal; patricia.rebelo@graq.isep.ipp.pt (P.R.); mgsps@isep.ipp.pt (I.S.); cmm@isep.ipp.pt (C.D.-M.); 2REQUIMTE, LAQV, Departamento de Química e Bioquímica, Faculdade de Ciências, Universidade do Porto, Rua do Campo Alegre, s/n, 4619-007 Porto, Portugal; ncordeir@fc.up.pt

**Keywords:** antidepressants, environmental pollution, molecularly imprinted polymer, screen-printed electrode, computational studies

## Abstract

A novel molecularly imprinted polymer (MIP) has been developed based on a simple and sustainable strategy for the selective determination of citalopram (CTL) using screen-printed carbon electrodes (SPCEs). The MIP layer was prepared by electrochemical in situ polymerization of the 3-amino-4 hydroxybenzoic acid (AHBA) functional monomer and CTL as a template molecule. To simulate the polymerization mixture and predict the most suitable ratio between the template and functional monomer, computational studies, namely molecular dynamics (MD) simulations, were carried out. During the experimental preparation process, essential parameters controlling the performance of the MIP sensor, including CTL:AHBA concentration, number of polymerization cycles, and square wave voltammetry (SWV) frequency were investigated and optimized. The electrochemical characteristics of the prepared MIP sensor were evaluated by both cyclic voltammetry (CV) and electrochemical impedance spectroscopy (EIS) techniques. Based on the optimal conditions, a linear electrochemical response of the sensor was obtained by SWV measurements from 0.1 to 1.25 µmol L^−1^ with a limit of detection (LOD) of 0.162 µmol L^−1^ (S/N = 3). Moreover, the MIP sensor revealed excellent CTL selectivity against very close analogues, as well as high imprinting factor of 22. Its applicability in spiked river water samples demonstrated its potential for adequate monitoring of CTL. This sensor offers a facile strategy to achieve portability while expressing a willingness to care for the environment.

## 1. Introduction

Antidepressant drug consumption has been rapidly growing every year. According to the Organization for Economic Cooperation and Development (OECD) statistics, the use of antidepressants doubled in 28 OECD countries between 2000 and 2017 [1,2]. Currently, their consumption levels are breaking records, and increased inputs of these compounds into environmental compartments have been reported by the scientific community [2,3,4,5,6,7]. In the future, some authors believe that this problem of emerging concern will be amplified as an immediate effect of the COVID-19 pandemic on the mental health of the communities [5,8,9].

Citalopram (CTL) is one of the most important antidepressants of the serotonin reuptake inhibitors (SSRIs) class, which is widely used throughout the world [5,10]. Its mechanism of action is based on blocking the reuptake of the neurotransmitter serotonin into nerve terminals of donor cells, thereby potentiating the serotonergic activity [11,12]. This compound is a racemic mixture of R(−) and S(+) enantiomers, but its desired pharmacological effect appears to reside mainly in the S-enantiomer [13,14]. In comparison with the previously introduced tricyclic antidepressants, CTL offers advantages owing to its therapeutic profile, superior acceptability, and safety, which justifies its popularity in the treatment of major depressive disorders [3,15,16]. In parallel, there is limited information on its environmental fate and toxicity, raising concerns regarding the potential effects on exposed wildlife [17,18,19]. Therefore, the monitoring of its presence in environmental compartments is imperative.

Chromatographic techniques are the most commonly used methods to detect CTL in biological and water samples [4,20,21,22]. On the other hand, their evident drawbacks, such as the need for complex and expensive instruments, the long analysis time, and the use of large volumes of toxic solvents have encouraged the scientific community to design new approaches for the detection of pharmaceuticals. While electrochemical methods have become an interesting alternative due to their simplicity, fast response, and low cost, only few studies based on these techniques have been reported to date for the determination of CTL [13,23,24,25,26,27,28,29,30].

To improve the selectivity and performance of electrochemical techniques, the use of molecularly imprinted polymers (MIPs) is one of the most interesting strategies. Their integration with electrochemical sensors has attracted extensive attention on environmental monitoring [31,32,33]. Moreover, MIP-based electrochemical sensors are one of the great candidates as detection and quantification tools, due to their chemical/mechanical stability, high selectivity and sensitivity, ease of preparation, and the possibility of in situ and real-time measurements [33].

Among the different approaches applied in MIP synthesis, electropolymerization provides a very simple and rapid method (one-step synthesis) to immobilize polymeric films on the precise location of the electrode surface with good adherence [34]. Notably, electropolymerization allows the achievement of a more sustainable MIP concept, as this method does not require a crosslinker and an initiator, which is conventionally used in the traditional preparation of MIPs [35]. At the same time, small sample/reagent volumes can be used, especially when the MIP is electropolymerized on the surface of the miniaturized screen-printed electrodes (SPEs) [36,37]. Currently, sustainable chemistry, also known as green chemistry, has been the focus of many research fields aiming to minimize the negative impact of chemical products and processes on the environment [38]. Reducing the quantity of organic solvents used as well as the waste generated is a key action for sustainable chemistry [39]. Following this direction, the literature shows few examples of environmentally-friendly practices, involving green technologies and alternative solvents, that are employed for MIP sensors development [40]. Therefore, computational studies, which are the most relevant tools to reduce the synthesis of novel MIPs by trial and error, have been used by researchers for preliminary exploration of molecular imprinting processes [41,42]. Appropriate simulations and computations have led to new methods for describing, predicting, and analyzing the complexity of molecular imprinting process without repeated laboratory experiments, an endeavor that saves time, energy, and the disposal of chemicals [42,43,44,45]. Researchers have demonstrated that computational studies based on molecular dynamics (MD) simulations can provide a realistic description of functional monomer-template interactions and, at the same time, explore the effects of different monomer ratios in the polymerization mixture [42,46].

In this research work, an attempt was made to develop a sustainable electrochemical MIP-based sensor for detection of CTL based on a simple, fast, and low-cost design. To the best of our knowledge, this is the first study that presents a voltammetric MIP sensor for the detection of this compound through screen-printed carbon electrode (SPCE) modification, which is suitable for point-of-care analysis. Phosphate buffer solution was used as a “green” solvent for electropolymerization of the CTL with the building and functional monomer 3-amino-4-hydroxybenzoic acid (AHBA), as well as solvent extraction and supporting electrolyte. By taking full advantage of the use of MD simulations, five MIP pre-polymerization mixtures were designed and simulated to predict the most efficient MIP formulation. Validation of these simulations was experimentally complemented for subsequent analyses of their electrochemical performance. The applicability of the sensor was tested in Portuguese river water samples, highlighting its ability to be used for in situ analysis through integration with portable devices. This investigation draws attention to the importance of rational approach on sensor development.

## 2. Materials and Methods

### 2.1. Reagents and Solutions

The 3-amino-4-hydroxybenzoic acid (AHBA, 97%), citalopram hydrochloride (CTL), potassium hexacyanoferrate (III), potassium hexacyanoferrate (II) trihydrate, fluoxetine (FLX), and carbamazepine (CBZ) were acquired from Sigma-Aldrich. Citalopram propionic acid (CTL-PA), citalopram N-oxide (CTL-NO), didemethylcitalopram (DD-CTL), and demethylcitalopram (D-CTL) were supplied by H. Lundbeck (Copenhagen, Denmark). Stock solutions of CTL (5 mmol L^−1^) were prepared in phosphate buffer solution (0.1 M, pH 7) and used to prepare less concentrated standards. Phosphate buffer solution was prepared with a mixture of KH_2_PO_4_ and K_2_HPO_4_ (Riedel-de-Haën). Ultra-pure water (with resistivity value of 18.2 MΩ∙cm) obtained from a Millipore (Simplicity 185) water purification system was used throughout the work. The water samples were collected from one Portuguese river. All of the chemicals were used as supplied.

### 2.2. Apparatus and Equipment

In this work, all of the electrochemical techniques were performed using SPCEs (DRP-110, Methrom, Spain). The configuration consists of a three-electrode arrangement printed on an alumina substrate with a circular carbon working electrode (WE, 4 mm ø), a carbon auxiliary electrode (CE), and a silver pseudo-reference electrode (RE). Cyclic voltammetry (CV) and square wave voltammetry (SWV) were carried out on a Metrohm Autolab PGSTAT 204 potentiostat/galvanostat controlled by NOVA 1.11 software. Electrochemical impedance spectroscopy (EIS) was conducted using a Metrohm Autolab PGSTAT 128 N potentiostat/galvanostat. This device was controlled by NOVA 1.6 software. To measure pH, a Crison pH meter (model micropH 92002) with a combined glass electrode was used. All of the experiments were conducted at room temperature.

### 2.3. Fabrication of the MIP-CTL Sensor and Electrochemical Analysis

To produce a homogeneous and thin MIP-CTL film, 40 µL of polymerization solution, containing 0.5 mmol L^−1^ CTL and 1.5 mmol L^−1^ AHBA in phosphate buffer, was gently dropped on the surface of the SPCE. Electropolymerization was performed by 10 consecutive CV cycles from −0.2 to 1.5 V at the scan rate of 100 mV s^−1^. Then, the electrodes were carefully washed with deionized water and air dried. To extract CTL molecules from the polymeric matrix, the modified SPCEs were subjected to 5 CV scans between −0.5 and 1.2 V at 100 mV s^−1^ in phosphate buffer. The non-imprinted polymer (NIP sensor) was prepared using the same procedure, but in the absence of CTL as a template in the electropolymerization process.

For rebinding of CTL molecules, the MIP sensor was incubated with 40 µL CTL solution for 15 min. Then, the electrodes were rinsed, and their electrochemical response was evaluated by SWV measurements in phosphate buffer over a potential interval from 0.4 to 1.2 V with a frequency of 20 Hz, a potential step of 5 mV s^−1^, and a pulse amplitude of 50 mV s^−1^. All of the measurements were performed in phosphate buffer as redox probe to study the electrochemical response of the prepared MIP sensors and were made in triplicate.

Under optimized conditions, CV and EIS experiments were employed to characterize the different steps of MIP-CTL construction. For this purpose, the current and resistance variations of 2.5 mmol L^−1^ [Fe(CN)_6_]^3−/4−^ prepared in 0.1 M KCl were evaluated. The CVs were scanned at 100 mV s^−1^ between −0.2 and 0.5 V. EIS assays were operated at a potential of 0.2 V over the frequency range of 0.1 to 10 Hz with 50 frequencies distributed logarithmically and a signal amplitude of 10 mV. EIS experimental data were acquired by Nyquist plots fitted based on Randles equivalent electrical circuit model.

### 2.4. Computational Setup

The MD study was performed in a similar manner to our previous works [43,44,45]. In short, all of the MD simulations were carried out with the GROMACS 5.1.4 software package [47]. For CTL and AHBA molecules, the general amber force field (GAFF) was used as received from the ANTECHAMBER module (including partial charges) [48]. During the experiments, the synthesis of this MIP occurs in phosphate buffer (see Section 2.3). For the computational study, all of the molecular species were placed in water to represent the solvent. Solvent water, based on literature reports [45,49,50], was selected for modelling by the TIP3P FF [51]. Several concentrations of functional monomer in pre-polymerization mixtures were investigated (see Section 3.1; Table 1) to identify the most favorable CTL:AHBA ratio. Therefore, the number of interaction sites in the systems varied from 33,150 for 1:1 template:monomer ratio to 37,650 for 1:6 ratio. Although the pharmacological effect of citalopram is associated with the S(+) enantiomer, in the most common commercial form both enantiomers are present in a racemic mixture. Therefore, we decided to emulate this composition in our simulations.

For the systems initiation, all of the necessary molecules were placed randomly into cubic simulation boxes using the PACKMOL code [52]. After energy minimization with the steepest-descent algorithm, two short pre-equilibrations in the canonical (for 100 ps at 100 K) and isothermal-isobaric (for 500 ps at 298 K) ensembles were performed. This was followed by an annealing procedure during 1 ns in the *NVT* ensemble, consisting of slowly heating the system up to 370 K and then slowly cooling it down to 298 K, which allowed us to overcome possible energy barriers. During all of the above-described MD steps, integration times of 0.5 fs were employed and no bond constraints were applied. Then, the equilibration (for 1 ns in *NpT*) and production (for 20 ns in *NVT*) followed at 298.15 K and 1 bar (only equilibration) conditions. At these two last stages, a time step of 1 fs was used and the bonds with hydrogens were constrained with the LINCS algorithm [53]. To maintain constant temperature, the Nosé–Hoover thermostat [54,55] was applied throughout the work with a coupling constant of 2.0 ps. Moreover, to maintain the pressure at a fixed level, where applicable, the Parrinello–Rahman barostat [56] was employed with a coupling constant of 8.0 ps. The leap-frog algorithm [57] was used for integrating the equations of motion. Periodic boundary conditions were applied in all directions with a cut-off of 1.2 nm set for both energy and pressure.

GROMACS in-built tools were employed to calculate the density, *d*, self-diffusion coefficients, *D*, radial distribution functions (RDFs), and hydrogen-bonding (H-bond) interactions. Self-diffusion coefficients were only evaluated after examining the attainment of a diffusive regime. To accomplish this, the parameter *β* was computed for each species and each simulation cell, as comprehensively described elsewhere [58,59,60,61]. The GRACE software was used for the numerical integration of RDFs for coordination numbers (CNs) assessment. When applicable, the error (i.e., deviation of a given simulated value from an experimental one) of a physical property was estimated as follows: (|*X*_exp_ − *X*_sim_|)/*X*_exp_ × 100%, where *X* is the property obtained experimentally, *X*_exp_, or computationally, *X*_sim_. A schematic diagram of the simulated species, with the atom numbering used throughout this work, is provided in Figure 1.

## 3. Results and Discussion

### 3.1. MD Simulations

During the preparation of a MIP sensor, one of the most important factors to consider is the interaction between the template and the functional monomer, as this significantly impacts the number of selective binding sites created on the polymeric matrix. In this context, finding the optimal molar ratio of template to the functional monomer is of great importance. Therefore, to evaluate the effect of the polymerization mixture components concentration as well as to gain a better insight into the possible interactions that occur between CTL and AHBA during MIP synthesis, MD simulations were carried out. Five different pre-polymerization mixtures were tested with varying concentrations of AHBA (Table 1).

To achieve high accuracy in reproducing experimental polymerization conditions, all of the mixtures were simulated using water as solvent. At the same time, CTL was considered as a racemic mixture during the design of the initial simulation box. In addition, as can be seen in Table 1, proportions of equal molecules of S(+) and R(−) enantiomers were included in the mixtures composition. Following preliminary tests, it was found that S(+) CTL and R(−) CTL do not present notable differences in local structure. Indeed, radial distribution functions, *g*(*r*), revealed virtually identical interactions of each CTL enantiomer with AHBA (see Appendix A), suggesting that the developed MIP sensor will not be able to distinguish between CTL enantiomers. Further analyses will help in the treatment of both CTL enantiomers.

#### 3.1.1. Density and Self-Diffusion Coefficients

To control the quality of the simulations and the employed FF models, the density and self-diffusion coefficients were evaluated and listed in Table 2.

As expected, the results showed a good correlation between the increase in the systems density with the increase in the AHBA molar fraction. Regarding the values of simulated self-diffusion coefficients, a similar diffusive behavior between CTL and AHBA can be noticed. The diffusion values of these molecules decline gradually from mixture 1 to mixture 5, with the exception of a slight increase registered in mixture 3. Taking into account that CTL needs to present enough mobility to allow its diffusion into and out of the MIP binding sites [62], mixtures 4 and 5 seem to show an unfavorable performance. In contrast, valuable results are obtained for both mixtures 1 and 3, as these have higher values of CTL diffusion coefficients. However, it is necessary to collect more data and evaluate other factors to infer the suitability of these mixtures to the experiment.

#### 3.1.2. Local Structure Analysis: RDFs and H-Bonds

RDFs were calculated to analyze the predictions for binding of CTL and AHBA on a molecular level in different pre-polymerization mixtures. Interactions between specific parts of these compounds were studied and discussed here only for the most active atoms (depicted in Figure 1), which are identified in a series of initial tests. The resulted RDFs are shown in Figure 2. All of the studied mixtures demonstrated favorable CTL-AHBA interactions, where the well-defined peaks can be observed at about 0.2 nm, suggesting the establishment of a hydrogen-bonding network.

Notably, AHBA displays a good affinity for CTL by the tertiary amine (N^1^, see Figure 1), the single oxygen-linked group (O^1^), as well as for the cyan group (N^2^) and fluor atom (F^1^) of the two different substituted benzene rings. In parallel, the extent of the interaction of AHBA is carried out by four hydrogens atoms, namely the two hydrogens of the amine group (H^4^ and H^5^), and the two hydrogens of hydroxyl groups (H^6^ and H^7^). Undoubtedly, the interactions observed between N^2^_CTL_ and H^4,5,6,7^_AHBA_ have an outstanding role in comparison with the other interactions. This finding is supported by the height of these characteristic peaks, which are more or less strengthened in different mixtures. However, following the results mentioned above, mixture 3 was judged to present the most balanced complex, which is supported by its defined peaks with a more homogeneous interaction distribution between CTL and AHBA molecules. Therefore, this can be explained by the fact that AHBA molecules are more uniformly distributed around CTL when compared with the other mixtures, which can be particularly advantageous to obtain a strong imprinting effect.

Based on these results, the extent of hydrogen-bonds (H-bonds) formation in each mixture was analyzed following the framework of Luzar and Chandler geometrical criteria [63], according to which a hydrogen bond can be considered if the distance between the oxygens of two different molecules is less than 3.5 Å and, simultaneously, the angle formed by O-H···O bond is less than 30°. Table 3 summarizes the total amount of H-bonds per mixture and the number of interactions per CTL molecules. A notable increase in H-bonds was registered with the increase in AHBA concentration, which tends to stabilize after mixture 3. This result is rationalized as a type of saturation of CTL-AHBA binding capacity. Due to the fact that hydrogen bonds have a key role in the imprinting process with direct influence on MIP efficacy and taking into account the performance of all the addressed pre-polymerization mixtures in terms of density, self-diffusion, and RDFs, it can be assumed that mixture 3 exhibits the most appropriate behavior. Consequently, the most promising ratio between CTL and AHBA with potential to create favorable imprinting sites during the electropolymerization process is 1:3.

### 3.2. Electrochemical Preparation of MIP-CTL Sensor and Its Recognition Abilities

In this work, the molecular imprinting of CTL molecules on SPCE surface was performed by simple electropolymerization of an optimized phosphate buffer solution containing 0.5 mmol L^−1^ CTL as template and 1.5 mmol L^−1^ AHBA as functional monomer. In addition to its efficient interaction with CTL, AHBA was employed due to its ability to provide a sustainable design of MIP, since this aromatic functional monomer has been considered as one candidate for bio-based monomers [64]. Moreover, it is highly soluble in common solvents and can be easily obtained by a microorganism (*Streptomyces griseus*), which promotes a more responsible mass-production process [64,65].

Figure 3A shows the resulting CV curves of electropolymerization process, which is performed by 10 consecutive cycles between −0.2 and 1.5 V. Here, a poly(AHBA) film with trapped CTL molecules was created. As seen from the figure, two oxidation peaks at about 0.3 and 0.98 V were registered. While the first peak can be attributed to the electrochemical oxidation of the AHBA at the SPCE surface, the second peak predicts the oxidation of the embedded CTL molecules during MIP film formation. Consequently, significant differences were observed between the MIP and NIP film growth (Figure 3B).

In addition to the scans of the present NIP film electro-generation, the electrochemical oxidation peaks of AHBA are more pronounced than MIP, i.e., the oxidation peaks at 0.3 and 0.8 V. However, no oxidation peak was detected at 0.98 V (Figure 3B). These findings should stem from the absence of CTL molecules that in the case of MIP synthesis triggered the amplification of the current response and did not allow for a complete visualization of the AHBA oxidation peak at 0.8 V, but rather, highlighted the oxidation peak of CTL molecules at 0.98 V.

Following the preparation of the polymeric film, the imprinting effects on sensor performance, including the incorporation of CTL molecules in the MIP film, as well as its elution and subsequent rebinding interaction, were evaluated by SWV measurements in phosphate buffer solution. The response profile on NIP and MIP-CTL SPCE recorded at a potential window from 0.4 to 1.2 V is shown in Figure 4. Therefore, a remarkable difference is noticeable in the electrochemical behavior of the resulting NIP and MIP film growth. As depicted in Figure 4A, a broad oxidation peak at about 0.8 V was exclusively observed in the MIP sensor, demonstrating that CTL molecules were successfully entrapped in the polymeric MIP layer deposited on the SPCE surface.

Commonly, to remove the trapped molecules from MIP and leave the rebinding cavities, solvent extraction is adopted. However, from a sustainable development perspective, this procedure was replaced by a fast and simple electrochemical overoxidation process. In this work, the extraction of CTL molecules from poly(AHBA) matrix was achieved by scanning the potential between −0.5 and 1.4 V in phosphate buffer solution, without the use of organic solvents. The resulting CV voltammograms (Appendix A) revealed that the increase in the cycle number displayed a decreasing peak current intensity until no obvious peak appeared after five cycles. This behavior suggests that CTL template molecules were removed from the imprinted polymer, as it was confirmed by SWV analysis, where no current peak was again observed (Figure 4B). Next, the rebinding of 1 µmol L^−1^ CTL showed a significant difference between the intensity of the obtained oxidation peaks from MIP and the corresponding NIP. The higher binding capacity of the MIP sensor is clearly attributed to the imprinting effect, which allowed it to adsorb a large amount of target molecules. Consequently, all of these outcomes prove the successful construction of MIP layer with complementary cavities to CTL molecules.

### 3.3. Optimization of Experimental Parameters in Preparation and Detection Process

The molar ratio between the template and functional monomer, the number of cycles during the electropolymerization process, and the frequency of SWV measurements are critical parameters that are responsible for the characteristics regulation and performance of the designed MIP sensor. To achieve the maximum current response and ensure an optimal selectivity of the sensor, the influence of these parameters on electrochemical response was evaluated in both MIP and NIP sensors. The results are illustrated in Figure 5.

Taking into consideration the results predicted by the described above MD simulations, the same conditions of template-functional monomer molar ratio were also experimentally studied to validate the accuracy of the proposed method. For this, the concentration of 0.5 mmol L^−1^ CTL remained constant and five different ratios of AHBA were tested (0.5, 1, 1.5, 2, and 3 mmol L^−1^), with varying ratios of CTL-AHBA. Notably, the NIP sensor was developed in the absence of CTL molecules and replaced by the phosphate buffer solution. As it can be seen in Figure 5A, the response signal of MIP sensor to CTL increases up to 1:3 ratio, suggesting that the higher amount of functional monomer provides more effective binding sites for CTL rebinding. However, when the molar ratio exceeded 1:3, a downward trend was observed in MIP sensor, which is possibly due to the presence of an excess amount of monomer that hinders the proper extraction and rebinding of CTL molecules. At the same time, no significant differences in the response signal of the NIP sensor were registered. However, at very high AHBA concentrations, there is a clear decrease in the difference between the peak current intensity of the MIP and NIP sensors and, consequently, a decrease in sensor sensitivity. These findings are in line with the predictions proposed by the computational studies and confirm the optimum ratio for the template and functional monomer of 1:3.

To investigate the effect of the polymeric film thickness and, consequently, the diffusion rate of CTL, the effect of the number of cycles applied during the electropolymerization procedure was explored under variations of 5, 10, 15, and 20 scan cycles (Figure 5B). As it is can be seen in Figure 5B, this parameter has a strong influence on the sensor performance. Curiously, there is a gradual decrease in the peak current intensity of the MIP sensor with the number of cycles, indicating that the diffusion of CTL molecules is impaired by thicker sensing films. On the other hand, for comparison with the response of the NIP sensor, it was found that five polymerization cycles were unable to produce enough and selective imprinted cavities, as the difference between the electrochemical response of NIP and MIP sensors toward CTL is very low. The highest difference was obtained by applying 10 scan cycles in the polymerization step. For this reason, 10 cycles were selected for further experiments.

To obtain the best analytical performance, the effect of the frequency parameter on SWV measurements was also studied by increasing the frequency from 10 to 50 Hz (Figure 5C). The results presented in Figure 5C show that with the increase in SWV frequencies, a significant enhancement in the electrochemical signal of MIP sensor until a maximum of 40 Hz occurs. In MIPs, the decrease in the current response at frequencies higher than 40 Hz can be the reflection of insufficient time for diffusion of CTL molecules within imprinted cavities, thus limiting the electrochemical activity. Considering the maximum difference between the responses of the NIP and MIP sensors, a frequency of 20 Hz was used in this work as an optimum value of SWV frequency. Additionally, the developed MIP sensor was subjected to time incubation studies. It was found that 15 min of incubation period was optimal for obtaining adequate rebinding of CTL on the MIP sensor.

### 3.4. Electrochemical Characterizations of the Stepwise MIP-CTL Construction

The successful construction of MIP-CTL sensor was monitored step by step, analyzing the changes in the electron transport properties of the sensing surface. For this purpose, CV and EIS measurements were performed in a 0.1 M KCl solution containing 2.5 mmol L^−1^ [Fe(CN)_6_]^3−/4−^ as redox probe. The electrochemical behavior of the bare SPCE and modified electrodes is shown in Figure 6.

As expected, the bare SPCE exhibits the highest redox characteristic peaks for [Fe(CN)_6_]^3−/4−^ in voltammogram (Figure 6A). When the poly(AHBA) film modified the SPCE surface, both NIP and MIP sensors registered a decrease in the redox peak current of [Fe(CN)_6_]^3−/4−^ as a result of the non-conductive polymer produced at SPCE surface that blocked the electron transfer. Distinctly, the MIP sensor shows a remarkable decrease in the peak current intensity, which is consistent with the entrapment of CTL in the polymeric film. The conductivity of the MIP sensor was recovered by extracting CTL molecules, which was accompanied by a significant increase in the peak current of the redox probe. Following incubation, the redox activity at MIP SPCE once again decreases, indicating that CTL occupied some of the previously formed cavities and, consequently impeded the ion diffusion of [Fe(CN)_6_]^3−/4−^ through the MIP layer.

The obtained information on the electrochemical characteristics of the electrodes after different modifications was complemented by EIS analyses. In accordance with CV data, the Nyquist plots displayed in Figure 6B showed significant differences between the impedance behavior of the electrodes. Following polymerization, a large increase in the diameter of the semicircle portion compared with the bare SPCE can be observed, implying an increase in the charge-transfer resistance (R_ct_). Therefore, the effective modification of the surface of SPCE was confirmed. Moreover, the R_ct_ presented by the MIP sensor was higher than the ones obtained with the NIP, which is ascribed to the presence of CTL acting as a barrier for [Fe(CN)_6_]^3−/4−^ diffusion. Furthermore, the removal of CTL molecules from MIP while leaving the channels free for the electron transfer process lowers the R_ct_. In contrast, when the MIP sensor was incubated, an enhancement in the diameter of the semicircle was again observed due to the obstruction effect. These observations prove the successful construction of the MIP sensor.

### 3.5. Analytical Performance

Under the optimized experimental conditions, the MIP sensor underwent direct SWV measurements for various concentrations of CTL (from 0.1 to 1.25 µmol L^−1^) in phosphate buffer (Figure 7). The increase in the oxidation current with the corresponding increase in the CTL concentration exhibited good linearity (*R*^2^ = 0.992, *n* = 3). The limit of detection (LOD, S/N = 3) and quantification (LOQ, S/N = 10) were found to be 0.162 and 0.541 µmol L^−1^, respectively.

### 3.6. Selectivity and Practical Application

Selectivity is the commendable feature of an effective MIP sensor. In this work, the selectivity of the developed sensor was probed by determining the current responses of five different compounds with similarity in chemical and spatial structure to the target CTL. Additionally, the electrochemical response of a tricyclic antidepressant was evaluated. For this purpose, both NIP and MIP sensors were separately exposed to CTL, CTL-PA, CTL-NO, DD-CTL, D-CTL, FLX, and CBZ under the same conditions (1 µmol L^−1^) for 15 min, and their corresponding SWV responses were compared. As illustrated in Figure 8, the prepared MIP sensor presents the highest sensitivity toward CTL molecules, while almost completely rejecting potential interfering analytes. The maximum imprinting factor (IF) is calculated as follows:(1)IF=ip(MIP)/ip(NIP)
where *ip* was as high as 22, corresponding to the peak current. No relevant analytical response was exhibited for CTL-PA, CTL-NO, DD-CTL, CTL-PA, and FLX. In addition, only a small electrochemical response was produced when the MIP sensor adsorbed these potential interferents, while no response was registered on their corresponding NIP sensors. The electrochemical response of MIP sensor to CBZ corresponds to 21% of the CTL oxidation peak, but there is a low significant difference between the intensity of MIP and NIP peak current. These findings reinforce the noticeable specific recognition of the sensor to CTL and confirm the success of the imprinting process without the need for toxic and hazardous chemicals.

Subsequently, the practical applicability of the prepared sensor for recognition of CTL in water samples was evaluated using the standard addition method. The recovery study was carried out by spiking the river water samples with known concentration of CTL (0.25 µmol L^−1^) and the analysis was performed without any pre-treatment. In these conditions, the recovery of this sensor was found to be 94.98%, indicating its sensing ability for determining CTL in water samples.

## 4. Conclusions

With hindsight on the design simplicity and unit activities offered by MIP implementation, the electrochemical MIP sensor developed in this work presents a valuable sustainable strategy to detect CTL. The adoption of MD simulations to optimize MIP formulation in a rational and greener method showed a good correlation with the experimental results, proving its power as a tool to improve the molecular imprinting process, while highlighting the need to explore this area to make the use of MIPs more eco-friendly. The successful assembly of MIP on SPCE surface as well as avoiding organic solvents in the whole analytical procedure, including the CTL extraction step and reducing waste production led to a good sensitivity, as revealed by LOD of 0.162 µmol L^−1^ and LOQ of 0.541 µmol L^−1^. Furthermore, the sensor showed an interesting recovery in spiked water samples, thereby demonstrating its applicability in water. Notably, the two figures of merit for the developed MIP sensor rely on its IF of 22 and, consequently, on its excellent selectivity toward CTL molecules. Of greater significance in environmental analysis, the developed MIP sensor reconciles the greenness of the method with potential functionality as a portable and low-cost sensor to CTL detection.

## Figures and Tables

**Figure 1 molecules-27-03315-f001:**
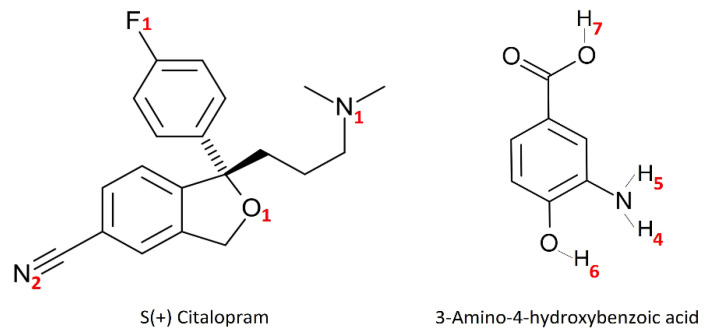
Schematic representation of S(+) citalopram, CTL with the chiral center indicated, and 3-amino-4-hydroxybenzoic acid (AHBA), along with some atom types, used throughout this work.

**Figure 2 molecules-27-03315-f002:**
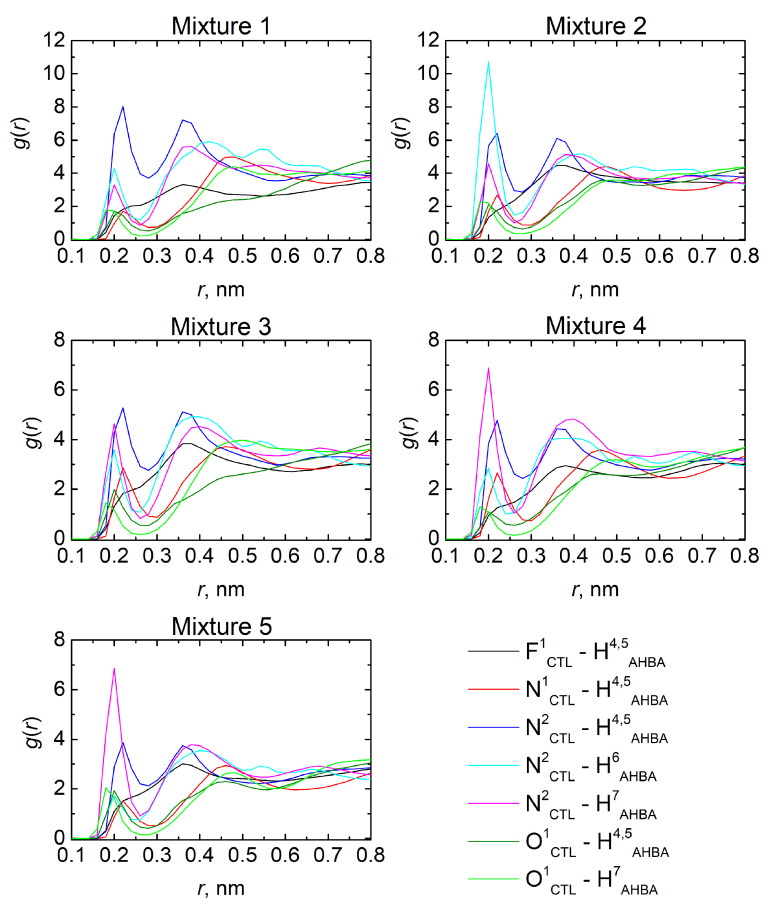
Radial distribution functions, *g(r),* between specific sites of CTL and AHBA molecules in the studied pre-polymerization mixtures from the results of molecular dynamics simulations.

**Figure 3 molecules-27-03315-f003:**
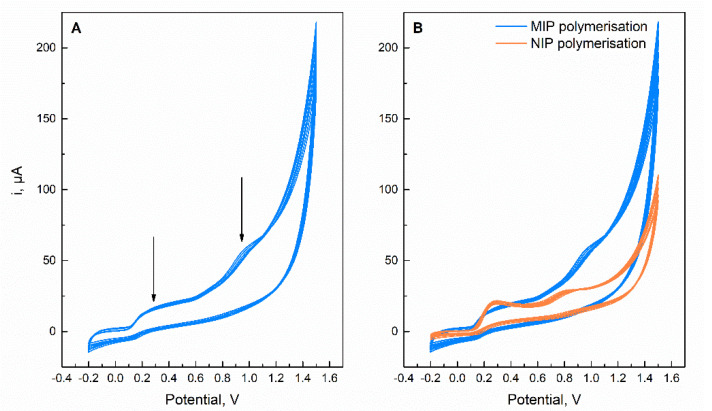
(**A**) Cyclic voltammograms of the MIP SPCE modification with 1.5 mmol L^−1^ ABA and 0.5 mmol L^−1^ ATV in buffer solution and (**B**) comparison of cyclic voltammograms building in MIP and NIP sensors.

**Figure 4 molecules-27-03315-f004:**
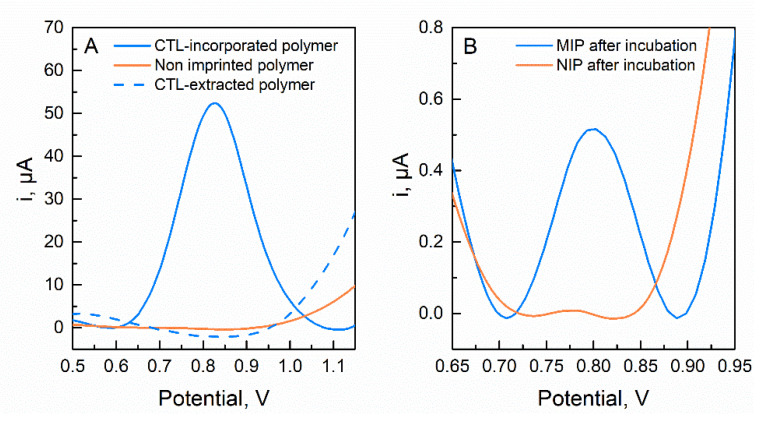
SWV voltammograms in 0.1 M of phosphate buffer (pH = 7) obtained after (**A**) MIP (0.5 mmol L^−1^ CTL, 1.5 mmol L^−1^ AHBA) and NIP (1.5 mmol L^−1^ AHBA) polymerization and extraction of CTL molecules; (**B**) MIP and NIP sensing responses after 15 min of 1 µmol L^−1^ CTL incubation.

**Figure 5 molecules-27-03315-f005:**
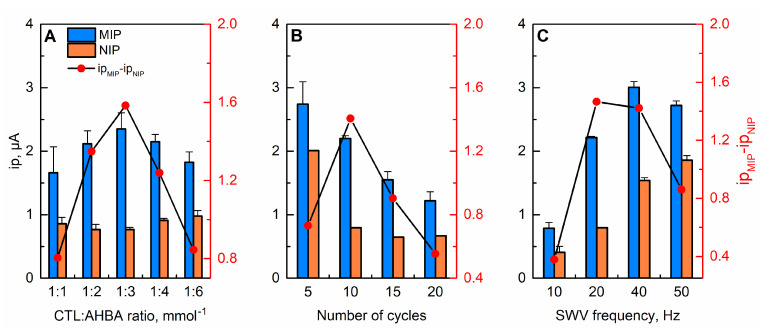
Effects of (**A**) molar ratio of CTL to AHBA, (**B**) number of polymerization scan cycles, and (**C**) frequency applied during SWV on sensing responses of MIP and NIP sensors toward 10 µmol L^−1^ CTL.

**Figure 6 molecules-27-03315-f006:**
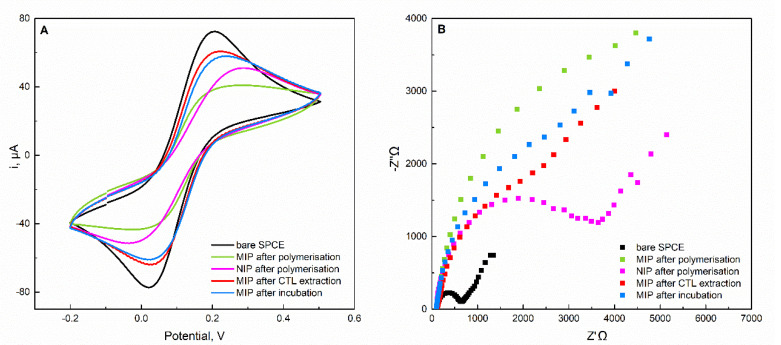
(**A**) Cyclic voltammograms and (**B**) impedance plots of a 2.5 mmol L^−1^ [Fe(CN)_6_]^3−/4−^ in 0.1 M KCl solution corresponding to the stepwise construction of the sensor.

**Figure 7 molecules-27-03315-f007:**
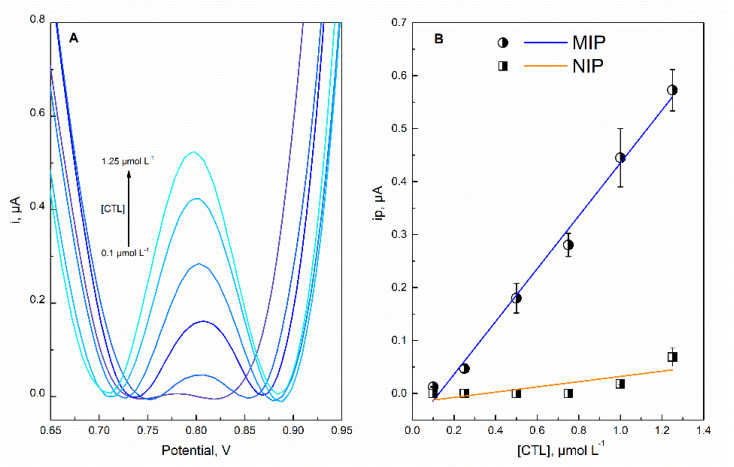
(**A**) SWV obtained from MIP response to CTL concentration in the range of 0.1–1.25 µmol L^−1^; (**B**) calibration plots for MIP and NIP sensors showing the relationship of CTL concentrations with the peak current intensity.

**Figure 8 molecules-27-03315-f008:**
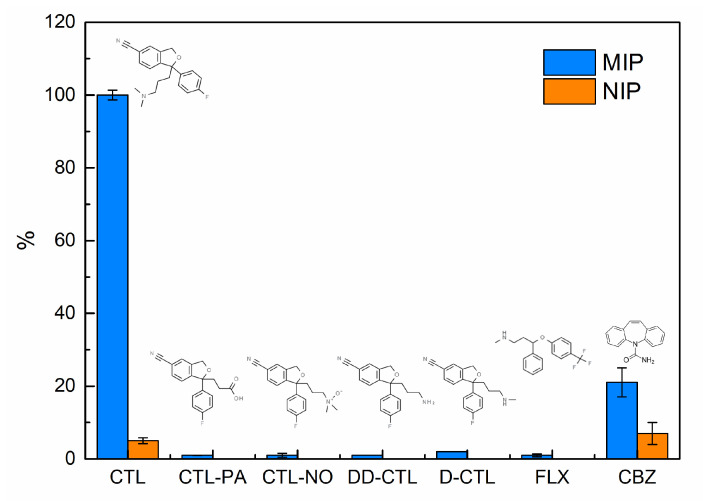
Electrochemical response of MIP and correspondent NIP sensor to potential interferent compounds with close structural similarities to CTL.

**Table 1 molecules-27-03315-t001:** Composition of the studied pre-polymerization mixtures.

		Number of Molecules				
Mixture	Molar Ratio (CTL:AHBA)	S(+)-CTL	R(−)-CTL	AHBA	H_2_O	Total Number of Interaction Sites
**1**	1:1	25	25	50	10,000	33,150
**2**	1:2	25	25	100	10,000	34,050
**3**	1:3	25	25	150	10,000	34,950
**4**	1:4	25	25	200	10,000	35,850
**5**	1:6	25	25	300	10,000	37,650

**Table 2 molecules-27-03315-t002:** Density (d) and self-diffusion coefficients (D) of the studied pre-polymerization mixtures.

		*D* × 10^9^, m^2^ s^−1^		
Mixture	d, kg m^−3^	CTL	AHBA	H_2_O
**1**	1009.43	0.13 ± 0.02	0.28 ± 0.01	5.21 ± 0.11
**2**	1021.88	0.11 ± 0.00	0.20 ± 0.02	4.87 ± 0.10
**3**	1032.86	0.12 ± 0.01	0.24 ± 0.02	4.53 ± 0.06
**4**	1043.96	0.03 ± 0.04	0.13 ± 0.00	4.32 ± 0.04
**5**	1064.54	0.06 ± 0.01	0.16 ± 0.03	3.94 ± 0.04

**Table 3 molecules-27-03315-t003:** Total average number of hydrogen bonds at different pre-polymerization compositions and their corresponding interactions as a function of CTL molecules.

Mixture	Total H-Bonds	Interactions *Per* 50 CTL Molecules
**1**	8.96	9
**2**	43.58	22
**3**	67.45	22
**4**	70.22	18
**5**	78.54	16

## Data Availability

Not available.

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
