# Peer review of "Computational Modelling and Sustainable Synthesis of a Highly Selective Electrochemical MIP-Based Sensor for Citalopram Detection"

_molecules, 2022, doi:10.3390/molecules27103315_

Round 1

Reviewer 1 Report

Although this work is interesting. The following concerns are observed. I recommend a major revision to address the queries. There are numerous MIP based electrochemical sensors for CTL and other similar analytes. The extraction process is not new either. The novelty of the process should be mentioned clearly.

CTL is an electroactive molecule. It can undergo direct electrochemistry with the electrode. What is the necessity for a MIP sensor here? If the selectivity is the concern, the author should show the response of CTL in the presence of other interfering species on bare electrodes and the response should be compared with the MIP sensor. The bar diagram should be represented with the error bars.

The results obtained using the MIP sensor for real-sample analysis should be correlated with the complementary methods and a difference should be discussed statistically.

The author claims that the oxidation peak observed at ~1.0 V corresponds to the oxidation of CTL molecules. However, the NIP formation curve (Figure 3B) recorded in the absence of CTL also shows an oxidation peak at ~0.85-0.9 V. This looks contractor and the understanding of the voltammetric signal requires more details.

The figure legend (Figure 4) was mentioned as MIP after polymerization. This does not make any sense. MIPs are polymers after template removal. So the legends should be modified as CTL incorporated polymer rather than MIP after polymerization. The same is the case with NIP after polymerization and MIP after CTL extraction.

The voltammetric current response of the MIP sensor system looks different than the CTL incorporated polymer-modified electrode, especially after 0.9 V. There is a steep increase in the current response of the MIP sensor system and there is no explanation for this. Also, the current output of the MIP sensor system is ~100 times lower than the CTL/polymer electrode. Again there is no explanation for such behaviour.

Author Response

Reviewer 1:

Although this work is interesting. The following concerns are observed. I recommend a major revision to address the queries. There are numerous MIP based electrochemical sensors for CTL and other similar analytes. The extraction process is not new either. The novelty of the process should be mentioned clearly.

We would like to acknowledge the reviewer’s time and his/her suggestions.

In fact, many biosensors have been constructed for the determination of CTL. However, to the best of our knowledge, this work reports for the first time on a development of a MIP-based voltammetric sensor to detect CTL. In the literature, to the best of our knowledge, we only find other MIP CTL sensor based on potentiometric readout and imprinted nanoparticles.

The electrochemical MIP sensor developed in this work presents a valuable sustainable strategy to detect CTL: computational simulations were employed to predict MIP formulation, and the imprinting process on SPCE surface were carried out avoiding the use of organic solvents in all analytical procedure, even in the CTL extraction step, following the concept of green analytical chemistry. Additionally, this investigation calls attention to the importance of rational approach on sensor development.

Now, we have included more information in the final paragraph of the introduction (line 91-104).

CTL is an electroactive molecule. It can undergo direct electrochemistry with the electrode. What is the necessity for a MIP sensor here? If the selectivity is the concern, the author should show the response of CTL in the presence of other interfering species on bare electrodes and the response should be compared with the MIP sensor. The bar diagram should be represented with the error bars.

We thank the reviewer for pointing this out. MIP sensors are currently the most versatile and cost-effective approach to create synthetic molecular receptors, offering advantageous properties like high affinity, stability and selectivity. MIP sensors has attracted attention in environmental monitoring. In the section “3.6 Selectivity and practical application”, namely figure 8, both NIP and MIP electrochemical response were compared with the response obtained towards interfering species (CTL-PA, CTL-NO, DD-CTL, D-CTL and FLX). However, no electrochemical response after incubation of CTL-PA, CTL-NO, DD-CTL, D-CTL and FLX was registered on bare electrodes. To make clearer, we have reformulated the phrase in the manuscript “No relevant analytical response was exhibited for CTL-PA, CTL-NO, DD-CTL, CTL-PA and FLX, and only a small electrochemical response was produced when the MIP sensor adsorbed these potential interferents, while no response was registered on their corresponding NIP sensors

 Additionally, according to the comment of the reviewer#2, we have included the electrochemical response of the carbamazepine (other class of antidepressants) to complete the selectivity of the proposed method.

Now, the bar diagram was represented with the error bars and we would apologize for the lapse.

The results obtained using the MIP sensor for real-sample analysis should be correlated with the complementary methods and a difference should be discussed statistically.

We appreciate the reviewer suggestion. The sensor was developed, and a proof of concept was tested using a drop of river water spiked with CTL without any treatment. This is one of the advantages of the use of electrochemical sensors as a detection approach. The comparison with a chromatographic method will be done in the future. At the moment we do not have the LC-MS methodology available in our group. However, the recovery study suggests a good possibility of application in environmental waters.

The author claims that the oxidation peak observed at ~1.0 V corresponds to the oxidation of CTL molecules. However, the NIP formation curve (Figure 3B) recorded in the absence of CTL also shows an oxidation peak at ~0.85-0.9 V. This looks contractor and the understanding of the voltammetric signal requires more details.

We thank the reviewer for the comment. In fact, The NIP formation curve presents an oxidation peak at ~0.85-0.9 V, which can be attributed to the oxidation of the functional monomer, 3-amino-4 hydroxybenzoic acid (AHBA). We verify that the functional monomer presents different oxidation peaks in the chosen experimental conditions, where the pH value of the medium (pH 7) was important to distinguish between AHBA and CTL oxidation peaks. At the same time, this difference between AHBA and CTL oxidation peaks is observed after polymerization (Figure 4A), where an oxidation peak was registered in MIP sensor, while no oxidation peak was observed in NIP sensor, confirming that the peak observed at about 1.0 V corresponds to the incorporation of CTL molecules. Nevertheless, we agree with the reviewer and we have completed the section “3.2. Electrochemical preparation of MIP-CTL sensor and its recognition abilities”:

“Besides the scans of NIP film electro-generation present electrochemical oxidation peaks of AHBA more pronounced than MIP, i.e. the oxidation peaks at 0.3 and 0.8 V, no oxidation peak at 0.98 V was detected (Figure 3B). These findings should stem from the absence of CTL molecules that in the case of MIP synthesis triggered the amplification of the current response, not allowing a complete visualization of the AHBA oxidation peak at 0.8 V, but instead highlighting the oxidation peak of CTL molecules at 0.98 V.”

The figure legend (Figure 4) was mentioned as MIP after polymerization. This does not make any sense. MIPs are polymers after template removal. So the legends should be modified as CTL incorporated polymer rather than MIP after polymerization. The same is the case with NIP after polymerization and MIP after CTL extraction.

We thank the reviewer for pointing this out. The legend of the figure 4 was modified as suggested.

The voltammetric current response of the MIP sensor system looks different than the CTL incorporated polymer-modified electrode, especially after 0.9 V. There is a steep increase in the current response of the MIP sensor system and there is no explanation for this. Also, the current output of the MIP sensor system is ~100 times lower than the CTL/polymer electrode. Again there is no explanation for such behaviour.

We understand the reviewer’s question. However, the same increase in the current happens in the CTL incorporated polymer-modified electrode after CTL oxidation peak as suggest by the figure 4A. The voltammetric current response is affected by the oxidation peak and the characteristics of the sensor. So, these variations are common in these systems. Considering that the polymerization and incubation were carried out with 0.5 mmol L-1 CTL and 0.001 mmol L-1 CTL, respectively (described in the legend), there is a significant difference in the intensity of the oxidation peaks, which justify this behaviour, as well as the decrease in the current output of the proposed MIP sensor (Figure 4B).

Reviewer 2 Report

The present work presents an interesting science and presents the development of a very systematic analytical methodology that involves a current and relevant subject.

However, some discussion is needed before publication in Molecules

1) I suggest that the authors include a study on the effect of the pH of the medium in relation to the performance of the proposed device

2) I also suggest that the effect of the ionic strength of the medium be evaluated

3) In fig 3b, what is the anodic peak in the region between 0.8 and 1.0 V attributed to in the electropolymerization process when NIP is used?

4) I suggest that other classes of antidepressants be evaluated in order to assess the selectivity of the proposed method

5) I suggest that the CTL determination be performed by the standard method and compared with the proposed method

6) I strongly suggest that the authors include a comparative table between the analytical parameters obtained by this work and those existing in the literature in order to determine CTL

Author Response

Reviewer 2:

The present work presents an interesting science and presents the development of a very systematic analytical methodology that involves a current and relevant subject.

However, some discussion is needed before publication in Molecules

We would like to acknowledge the reviewer’s time and his/her suggestions.

1) I suggest that the authors include a study on the effect of the pH of the medium in relation to the performance of the proposed device

We appreciate the reviewer suggestion. In fact, pH value is one of the important factors that can affect the electrochemical performance of the sensor and, accordingly, we made a study about its influence on efficiency of the proposed MIP sensor. However, considering that the oxidation peak of CTL can be shifted to less or more positive potentials in accordance with the value of pH, we find some difficulties to distinguish the oxidation peaks of CTL and AHBA. So, pH 7 was selected as optimum pH for the MIP construction and the supporting electrolyte for sensitive determinations.

2) I also suggest that the effect of the ionic strength of the medium be evaluated

We appreciate the reviewer suggestion; however, it is not a common study for this kind of MIP electrochemical sensors and in our point of view, the study of the effect of the ionic strength of the medium is not very important for the sensor performance and selectivity.

3) In fig 3b, what is the anodic peak in the region between 0.8 and 1.0 V attributed to in the electropolymerization process when NIP is used?

The NIP formation curve presents an oxidation peak at ~0.85-0.9 V, which can be attributed to the oxidation of the functional monomer, 3-amino-4 hydroxybenzoic acid (AHBA). To make this clearer, we have added this information in the section “3.2. Electrochemical preparation of MIP-CTL sensor and its recognition abilities”: Besides the scans of NIP film electro-generation present electrochemical oxidation peaks of AHBA more pronounced than MIP, i.e. the oxidation peaks at 0.3 and 0.8 V, no oxidation peak at 0.98 V was detected (Figure 3B).

4) I suggest that other classes of antidepressants be evaluated in order to assess the selectivity of the proposed method.

We thank the reviewer for this suggestion. Now, we have included in the figure 8 the electrochemical response of MIP and correspondent NIP sensor to carbamazepine (CBZ), the only available tricyclic antidepressant in our laboratory. Accordingly, the section “3.6 Selectivity and practical application” has been rewritten.

5) I suggest that the CTL determination be performed by the standard method and compared with the proposed method

We appreciate the reviewer suggestion. The sensor was developed, and a proof of concept was tested using a drop of river water spiked with CTL without any treatment. This is one of the advantages of the use of electrochemical sensors as a detection approach. The comparison with a chromatographic method will be done in the future. At the moment we do not have the LC-MS methodology available in our group. However, the recovery study suggests a good possibility of application in environmental waters.

6) I strongly suggest that the authors include a comparative table between the analytical parameters obtained by this work and those existing in the literature in order to determine CTL.

We thank the reviewer for pointing this out. However, we did not include a comparative table because, to the best of our knowledge, only other CTL sensor based on potentiometric readout and imprinted nanoparticles with a comparable limit of detection (0.125 µmol L-1) was reported. At the same time, the sensor was used to measure CTL in urine. The practical applicability of the other electrochemical sensors has been tested in human biological fluids and pharmaceutical samples, using higher concentrations in a real applicability. The determination of pharmaceuticals in environmental water still remains a challenge.

Reviewer 3 Report

The manuscript reports the MIP sensor used to detect citalopram, corroborated by the computational modeling (MD). The experimental and computational method and is described clearly. The discussion of the result is adequate. 

I suggest publication of the manuscript as is.

Author Response

We would like to acknowledge the reviewer’s time

Round 2

Reviewer 2 Report

The suggestions were not fully accepted.

This manuscript is a resubmission of an earlier submission. The following is a list of the peer review reports and author responses from that submission.

Round 1

Reviewer 1 Report

This manuscript describes about computational modeling, molecular imprinting and electrochemical sensor.

Their target is citalopram (CTL) which is an antidepressant drug. CTL is not a biomolecule. The limit of detection (LOD) is either 0.162 mmol L-1 (Ab) or 0.162 mol L-1 (Conclusion), which is not comparable to other bio-detection (nmol L-1). Therefore, it is not belong to the scope of biosensor. I would suggest the authors considering other MDPI journal, like sensors or chemosensors.

The technique they used in molecular imprinting and electrochemical sensor is not new. It has been reported. (Sensors and Actuators B: Chemical, 2017, 243, 745-752.)

No distinctive advantage was disclosed in this MS about using computational modeling for optimizing molecular imprinting. According to Fig 5, the difference between MIP and NIP are not much. Suddenly, MIP and NIP are significant in Fig 8.

The formation of CTL-AHBA to CTL-MIP should be drawn.

What is the electrochemical reaction of CTL?

Several typo errors were found. 3-amino-4-hydroxybenzoic acid (AHBA) Fig 3 and Fig 5 ABA?

Reviewer 2 Report

This manuscript reports on electrochemical molecularly imprinted polymer (MIP), presents a strategy to detect citalopram (CTL) in environmental waters. The adoption of MD simulations optimize MIP formulation in rational. The electrochemical characteristics of the prepared MIP sensor were evaluated. The limit of detection (LOD) of 0.162 mmol L-1 (S/N=3). And its applicability in spiked river water samples demonstrated. I listed my comments below:

  1. The full name of SWV should written in the abstract.
  2. The MIP layer was prepared by electrochemical in-situ polymerisation of the 3-amino-4- hydroxybenzoic acid (AHBA) functional monomer. What is the author's basis for choosing this functional monomer?
  3. The further explanation of the optimal conditions is needed for Figure 5.
  4. The authors claim that hydrogen bonds have a key role in the imprinting process with a direct influence on MIP efficacy. Does the pH value have an effect on its efficiency?
  5. Is it more intuitive to place the structural formulas of different compounds in Figure 8?

Reviewer 3 Report

The authors presented a nice work in the determination of citalopram (CTL) using screen printed carbon electrodes. MD simulations were used to optimize MIP formulation, and a good correlation with experimental results was obtained. I recommend publishing after minor revision.

(1) SWV was not defined before its first usage in the Abstract.

(2) Introduction, especially Paragraphs 1 and 2 need to be revised to be concise.

(3) Page 2, line 88-89

The authors claim that “…computational studies…have become very popular” while only two papers were cited. The authors need to revise the sentence to tell readers whether the top is really that popular.

(4) Page2, line 93-96

Proper literature should be cited.

(5) Figure 1

The scheme for 3-Amino-4-hydroxybenzoic acid is not correct.

(6) Table 2

As shown in Table 2, both CTL and AHBA of mixture 4 showed the lowest self-diffusion coefficients. Can the authors explain why they were not following the declining trend from mixture 1 to 5?

(7) Page 8, line 296-298

It’s hard to understand “A notable increase 296 of H-bonds was registered with the increase of AHBA concentration, which tends to stabilise after mixture 2 (considering the total number of H-bonds) or mixture 3 (considering the number of interactions per 50 CTL molecules)”.

(8) Figure 3

Please list the reference electrode in Figure 3. Figure 3B should also show the CV of MIP and NIP after consecutive cycles test. There is no discussion on Figure 3B.

(9) Figure 4

The Y-axis for Figure 4b is missed. Is the scale correct?

(10) Table 4

Generally, the work showed no obvious advantage over other methods listed in Table 4 considering the linear range and LOD. Do the authors have a plan to optimize the strategy? And it is better to be added to the discussion.

(11) Page 3 line 102-106

Page 4, line 187

Page 14, line 578-580

Reviewer 4 Report

The manuscript submitted by P. Rebelo and collaborators may be of interest to Biosensors readers, but there are some important issues that need to be clarified or remedied by the authors, which are listed below:
- The detection of citalopram in water samples (environmental application) is of interest, but it is not clear whether the sensor developed by the authors is suitable for the proposed purpose. Clearly, the level of citalopram in water (including wastewater is very low), but this information is missing from the manuscript. The authors are asked to include in the manuscript a paragraph in which to discuss the practical applicability of the sensor with reference to the drug level in the water. If the analytical performance of the sensor is not good enough to be able to directly detect the target analyte in water samples, what are the solutions proposed by the authors for testing the real samples?
- I noticed that the PIN control sensor was developed by the authors. However, the data obtained by the detection method in the case of the PIN sensor tested after contact with different concentrations of the drug are not entered in the manuscript. Usually, in the case of MIP sensors, the complementary NIP sensor is also tested to establish the influence of nonspecific adsorption in the analytical signal. This is not the case here, and the authors are asked to represent the results compared to those on MIP, in Figure 7B

  • Authors do not demonstrate real imprinting of the prepared layer; all results presented might be obtained also from non-specific adsorption. And this is the case already claimed in the field, that just showing a signal gain, or an imprinting factor, is not sufficient for demonstrating an imprinting effect (see recommendations in J. Molec. Recognition (2011): 1115-1122). The absolute demonstration of these claims resides in finding the thermodynamical constants of the complex formation with the analyte, and doing a proper comparison between MIP and NIP. If subjacent mechanisms are or are not non-specific adsorption, this may be demonstrated after the binding analysis of the generated material. The obtaining and interpretation of such binding isotherms are then the only transparent proof that imprinting is in the basis of the operation of the generated recognition layer.
  • The entire manuscript must be carefully revised because there are errors in wording, expression and language
  • Only with these claims solved in the revised paper, I can recommend publication.

Reviewer 5 Report

The authors of this proposed paper have developed a sensor for the selective detection of the antidepressant citalopram. The authors have optimized the receptor layers in their sensor by computational modeling and optimization of the electrochemical deposition process. Although detection of antidepressants by biosensors and MIP-based sensor platform in itself is not really novel, I feel the approach is sufficiently different to fit the scope and aims of MDPI Biosensors. The work is also of sufficient quality but I feel some additional crucial experimental data should be added and the document could use a little more elaboration on certain aspects of the sensor before publication in this journal is warranted. Therefore, I would advise to address the comments raised below:

  1. I am a bit confused by the opening sentence of the abstract. What do you mean by "the first sensor"? Many biosensors have been constructed for this compound over the years with many of them based on electrochemical detection.
  2. The authors nicely show how they optimized the ratio between functional monomers and the target through hydrogen bonding. This rational design approach seems logical and I feel it is not often enough explored in MIP sensor design. However, I was wondering if the authors also considered other functional monomers, did you do similar simulations to determine AHBA as the ideal monomer? Or was the choice for AHBA made prior to doing the simulations purely on the base of its structure and potential to electropolymerise?
  3. The authors mention that their MIP system will not be able to distinguish between different enantiomers of CTL. Some other studies do surprisingly show enantioselectivity. Do the authors see this as a drawback of their sensor or does it also pose some advantages? Did you also do some experiments or did you only base yourself on the simulations? Please elaborate a bit on this when discussing it in the text as I feel it is an important characteristic of the sensor.
  4. I was a bit surprised that the authors again vary the molar ratios of CTL and AHBA during the optimization of the electropolymerisation procedure, while I felt that you already simulated them in the MD experiments. You do briefly mention you do it again but you do not really compare the results of both. Also the authors mention a downward trend after 1:3 but are the differences statistically relevant in terms of the error on the data? At first glance the differences seem small?
  5. The main point that should be improved in the paper is the core characterization of the sensor itself. You basically only characterize it in very narrow concentration range in Figure 7 spanning only 1 order of magnitude. The lower end of this range seems fine because your LoD falls within it. But the higher range is not necessarily the end of your linear range? What happens if you increase the concentration? Will the signal saturate or at least not follow a linear regime anymore? I feel these measurements should be added to reliably compare your performance to those of other sensors summarized in Table 4.
  6. The comparison to Table 4 should be much more elaborated on in general. You basically only mention that your straightforward approach is competitive... But is it? Apart from 1 sensor, I feel your sensor's LoD is 1-2 orders of magnitude less and your linear range only spans one order of magnitude while the other sensors at least span two. Is this essential for the proper use of the sensor? What type of concentrations do you expert in real life? This needs to be studied and implemented into your discussion. 

Round 2

Reviewer 1 Report

The grammar and spelling are much better in the revised version.

However, the sensitivity of MIP electrochemical sensor is still the next to last in Table 4. Although the authors claimed the effect of computer modeling, no distinctive effect on their sensitivity was improved in Table 4.

The authors explained the MIP/NIP ratio of Fig 4, 5 and 8, but I am still in the mist.

Fig 4 and 8 are matched using 1 μmol L-1 CTL but Fig 5 is not. Why figure 5 were performed using 10 μmol L-1 CTL? Line 439, the authors make a clear sum-up for Fig 5. Is this true? To avoid confusion, please draw figure 5 using 1 μmol L-1 CTL.

BTW, what is the thickness of the polymeric film?

Reviewer 4 Report

I read the authors' answers and found that they did not take into account the observations and modified the manuscript very little. The changes made are mainly in terms of language and very little as discussion and comparison of the study with literature data.
With all respect to the authors, I maintain my observations from the first round of review, and in this form, I cannot recommend the publication of the study in Biosensors journal.

Reviewer 5 Report

The authors did address some of my comments but essentially added no experimental work. Two remarks therefore remain intact, one is a detail and one will require some additional experiments:

  1. I did a quick google scholar search and found a CTL sensor based on potentiometric readout and imprinted nanoparticles. So I still think you are not the first to introduce an electrochemical MIP sensor for it. A detail obviously but nevertheless you mention it 2-3 times in your article. 
  2. More importantly, I strongly disagree with the rebuttal on Figure 7 as your explanation does not make sense. You mentioned that you are only interested in trace levels. Yet you measure in the micromolar range while you yourself answer in your rebuttal to the next remark that the concentration range actually spans from ng/L to 76 microgram/L. The latter would be just below what you actually encounter. But you do not reach the low levels. But more importantly... it spans 3-4 orders of magnitude. So measure extra concentrations in higher regimes to show how broad your linear range is. This would allow you to discuss in a critical manner that if you would be able to bring down the sensitivity a couple of orders of magnitude you might actually have a useful tool for the future. 
